# CD34+ Stromal Cells/Telocytes as a Source of Cancer-Associated Fibroblasts (CAFs) in Invasive Lobular Carcinoma of the Breast

**DOI:** 10.3390/ijms22073686

**Published:** 2021-04-01

**Authors:** Lucio Díaz-Flores, Ricardo Gutiérrez, Miriam González-Gómez, Maria Pino García, Lucio Díaz-Flores, José Luís Carrasco, Pablo Martín-Vasallo

**Affiliations:** 1Department of Basic Medical Sciences, Faculty of Medicine, University of La Laguna, 38071 Tenerife, Spain; histology54@gmail.com (R.G.); mirgon@ull.es (M.G.-G.); ldfvmri@yahoo.com (L.D.-F.J.); jcarraju@ull.edu.es (J.L.C.); 2Department of Pathology, Eurofins Megalab–Hospiten Hospitals, 38100 Tenerife, Spain; mpgarcias@megalab.es; 3Developmental Biology Laboratory, Department of Biochemistry and Molecular Biology, University of La Laguna, 38100 Tenerife, Spain; pmartin@ull.edu.es

**Keywords:** CD34+ stromal cells, telocytes, cancer-associated fibroblasts, CAFs, invasive lobular carcinoma, breast

## Abstract

Several origins have been proposed for cancer-associated fibroblasts (CAFs), including resident CD34+ stromal cells/telocytes (CD34+SCs/TCs). The characteristics and arrangement of mammary CD34+SCs/TCs are well known and invasive lobular carcinoma of the breast (ILC) is one of the few malignant epithelial tumours with stromal cells that can express CD34 or αSMA, which could facilitate tracking these cells. Our objective is to assess whether tissue-resident CD34+SCs/TCs participate in the origin of CAFs in ILCs. For this purpose, using conventional and immunohistochemical procedures, we studied stromal cells in ILCs (n:42) and in normal breasts (n:6, also using electron microscopy). The results showed (a) the presence of anti-CD34+ or anti-αSMA+ stromal cells in varying proportion (from very rare in one of the markers to balanced) around nests/strands of neoplastic cells, (b) a similar arrangement and location of stromal cells in ILC to CD34+SCs/TCs in the normal breast, (c) both types of stromal cells coinciding around the same nest of neoplastic cells and (d) the coexpression of CD34 and αSMA in stromal cells in ILC. In conclusion, our findings support the hypothesis that resident CD34+SCs/TCs participate as an important source of CAFs in ILC. Further studies are required in this regard in other tumours.

## 1. Introduction

CD34+ stromal cells (CD34+SCs) are present in the connective tissue of most organs. A new cellular type, identified by electron microscopy and named telocyte (TC) [1,2] largely corresponds to the CD34+SCs observed in light microscopy (CD34+SCs/TCs) [1,3,4,5]. These cells show a small body and long cytoplasmic processes (telopodes), in which a velamentous morphology has been observed in 3D imaging by FIB-SEM tomography [6,7,8].

CD34+SCs/TCs are not observed in the stroma of numerous malignant invasive epithelial tumours, where they are replaced by αSMA+ stromal cells (myofibroblasts, cancer-associated fibroblasts: CAFs) [9,10,11,12,13,14,15,16,17,18,19,20,21,22]. Several origins have been proposed for CAFs, including tissue-resident CD34+SCs/TCs, bone marrow derived fibrocytes, pericytes and other possible precursor cells, such as endothelial cells (ECs), smooth muscle cells, adipocytes, and epithelial cells, to which the ability to transition or transdifferentiate has been attributed [23,24,25,26,27,28,29]. Among these cells, the participation of tissue-resident CD34+SCs/TCs as precursors of CAFs is very likely, as the process requiring the cellular loss of CD34 expression and the gain of αSMA expression.

The loss and gain of CD34 and αSMA expression, respectively, occurs in repair through granulation tissue [30,31,32]. In the early stages of human granulation tissue formation, we have observed changes from resident CD34+SCs/TCs to stromal cells expressing αSMA, including the transient coexpression of both markers [31,32,33]. These observations point to human tissue-resident CD34+SCs as an important source of myofibroblasts during repair [31,32,33]. In addition, CD34+SCs/TCs may be activated with or without myofibroblast transformation (e.g., depending on distance from lesion location) and with the consequent loss or persistence of CD34 expression. CD34+SC/TC evolution in αSMA+ cells in early stages of repair through granulation tissue can be demonstrated by controlling times, following the evolutive process and observing the coexpression of markers. However, this is not the case in neoplasms.

CD34 expression may be preserved in the stromal cells of some epithelial tumours, as occurs in lobular carcinoma of the breast [34]. In addition, the characteristics, distribution and behaviour of CD34+SCs/TCs in the normal breast have been well established [3,35,36,37,38,39,40,41]. A recent study of invasive lobular carcinoma confirmed the partial preservation of expression in tumour stromal cells, with varying presentation of CD34+ and αSMA+ stromal cells [42]. Thus, each type of stromal cell could be absent or present (focally, predominantly or uniformly) in the tumour [42]. Both types of stromal cells can therefore coincide in the same tumour. Findings, including coexpression of markers supporting the transformation of CD34+SCs into αSMA+ stromal cells, could be more easily observable in these tumours with both types of stromal cells around the nest and strands of neoplastic cells. A study of this possibility to confirm resident CD34+SC/TC participation as precursors of stromal cells (CAFs) in tumours would be of interest.

Given the above, the objective of this study is to assess whether resident CD34+SCs participate as precursors of stromal cells in tumours. To this end, we explored invasive lobular carcinomas of the breast to determine the following: (a) whether the pattern of arrangement and location of tumoral stromal cells (CD34+ and αSMA+ stromal cells) in the tumours coincide with the arrangement and location of CD34+SCs/TCs in the normal breast, (b) whether both types of stromal cells are seen around the same nest of neoplastic cells or even around a single neoplastic cell and, more importantly, (c) whether stromal cells co-expressing CD34 and αSMA are present around neoplastic cells in the tumours.

## 2. Results

### 2.1. Stromal Cells around Strands and Nests of Neoplastic Cells in Invasive Lobular Carcinoma

In all or broad areas of invasive lobular carcinomas, the solid nests and strands of neoplastic cells were surrounded by stromal cells, mainly by the long, thin processes emitted by them. Using double labelling in immunochemistry and immunofluorescence for CD34 and αSMA, only stromal cells expressing CD34 were seen in the normal breast. In the invasive lobular carcinoma (Table 1), we observed nests of neoplastic cells surrounded by anti-CD34+ stromal cells (Figure 1A) or by anti-αSMA+ stromal cells (Figure 1B). Both types of nests were also observed in proximity (Figure 1C). Each cell type was present in varying degrees, ranging from very rare to balanced. Thus, the cases were grouped into (a) predominance of CD34+ stromal cells (n:11, 26.19%), (b) balanced presence of both cell types (n:18, 42.85%) and (c) predominance of αSMA+ stromal cells (n:13, 30.9%) (Table 1).

ECs, which also expressed CD34, showed more intense staining with this marker than stromal cells around the neoplastic cells (Figure 2A). αSMA+ mural cells (pericytes or vascular smooth muscle cells) were frequently observed around ECs, even in smaller vessels when sectioned tangentially or with virtual lumen (inserts of Figure 2A). Anti-CD34+ stromal cells around neoplastic cells were anti-CD31 negative, while ECs showed coexpression of CD34 and CD31 (Figure 2B,C). The neoplastic cells presented expression for cytokeratin AE1/AE3 (Figure 2D,E).

CD34+SC/TC processes/telopodes were generally long and thin in the normal breast and in the stromal cells in tumours, whereas occasional observations in sections parallel to the surface of these processes had a flattened appearance. Thus, in invasive lobular carcinoma, depending on the section, αSMA+ stromal cells appear to form a veil around neoplastic cells (Figure 3A,B). This fact was also demonstrated using tissue sections of 10 µm in confocal microscopy by individual visions observed at high magnification and at different heights (Figure 3B,C).

### 2.2. The Patterns of Arrangement and Location of Stromal Cells (CD34+ and αSMA+ Stromal Cells) in Invasive Lobular Carcinoma Are Similar to Those of CD34+SCs/TCs in the Normal Breast

In the breast, CD34+SCs/TCs were more concentrated around the ducts and terminal ductal-lobular units, where they were arranged in several concentric layers (Figure 3E,F). The innermost CD34+SCs embraced the myointimal cells of the ducts and ductules (Figure 3F). Ultrastructurally, the stromal cells showed long cytoplasmic processes (telopodes) (Figure 3G,H), homocellular contacts (Figure 3G, insert), and podoms and podomeres (Figure 3H). These cells were separated from myointimal cells by the basement membrane of the latter and a collagenous band of varying thickness (Figure 3G,H).

In invasive lobular carcinoma, the stromal cells were seen in the same pattern of arrangement (concentric layers) and location as in normal conditions, but around neoplastic cells, which also acquired a concentric ring pattern (compare Figure 3E,F and Figure 5A with Figure 4 and Figure 5B,C). Neoplastic cell distribution in concentric rings (following the arrangement of the surrounding stromal cells) around terminal ducts reflected the pattern that is virtually diagnostic of invasive lobular carcinoma when studied by the routine haematoxylin and eosin technique (insert in Figure 4A). Double immunochemistry (Figure 4B) and immunofluorescence (Figure 5) labelling also revealed variations in CD34+ and αSMA+ stromal cell numbers in these locations around ducts. Thus, a predominance of CD34+ stromal cells (Figure 4B and Figure 5B), a balanced proportion of both types of cells and a predominance of αSMA+ cells (Figure 5C) were observed in this location around the ducts.

CD34+SC/TC distribution in the intralobular interstitium (Figure 6A) also corresponded to that of stromal cells around infiltrative neoplastic cells in invasive lobular carcinoma (Figure 6B).

CD34+SCs/TCs were also concentrated around blood vessels (Figure 6C). In this location, the quantity and number of layers of CD34+SCs/TCs depended on vessel size. For example, in arteries, there were numerous layers of CD34+SCs/TCs (Figure 6C). The same distribution was observed in stromal cells around neoplastic cells in invasive lobular carcinoma (Figure 6D). In small vessels, there was only one external layer of CD34+SCs/TCs (Figure 7A), and neoplastic cells were surrounded by processes of stromal cells arranged in one layer around small vessels in invasive lobular carcinoma (Figure 7B).

In the interlobular connective tissue of the breast, CD34+SCs/TCs had long processes, which were generally bipolar and were present around and parallel to collagen fibres. Ultrastructurally, these cells showed long, thin telopodes, which established homocellular contacts (Figure 7C). In invasive lobular carcinoma, stromal cells continued in the same location and their long processes were arranged around neoplastic cells, which appeared independently (Figure 7D) or frequently formed strands and nests.

In adipose tissue, CD34+SCs/TCs were observed between adipocytes (Figure 7E) in the connective interlobular tracts and in the submammary region. This distribution was preserved around neoplastic cell nests in invasive lobular carcinoma in the adipose tissue (Figure 7F).

In the interstitium of the smooth muscle of the mammary nipple, CD34+SC/TC distribution also corresponded to that around neoplastic cells in this location (Figure 8A,B). αSMA+ stromal cells were also observed in the epi-perineurium surrounding neoplastic cells in nerves, while the endoneurial cells conserved CD34 expression (Figure 8C).

Therefore, the data set show that the location and arrangement of stromal cells in invasive lobular carcinoma coincide with those of CD34+SCs/TCs in the normal breast (Table 2).

### 2.3. Presence of Both Types of Stromal Cells (CD34+ and αSMA+ Stromal Cells) around the Same Strand and Nest of Neoplastic Cells

In cases presenting both types of stromal cells, double immunochemistry for anti-CD34 and anti-αSMA revealed CD34+SCs and αSMA+ stromal cells around the single nests (Figure 8D) and strands of neoplastic cells (Figure 8E and Table 2). Observations using this procedure suggested coexpression of CD34 and αSMA in some of these stromal cells (see below).

### 2.4. Coexpression of CD34 and αSMA in Stromal Cells of Invasive Lobular Carcinoma

Coexpression of CD34 and αSMA in stromal cells surrounding neoplastic cells was demonstrated by double labelling fluorescent confocal microscopic analysis of these markers (Figure 9 and Figure 10). Semiquantitative analysis showed that the number of cells co-expressing CD34 and αSMA varied depending on the tumour area examined (Figure 9 and Figure 10). Thus, in the areas with the highest coexpression, the percentage of stromal cells co-expressing both markers was greater than 25% for all the groups of invasive lobular carcinoma, with a higher incidence in the groups with a predominance of CD34+ stromal cells and a balanced proportion of the two types of cells (Table 2).

In some nests and strands of neoplastic cells with coexpression of CD34 and αSMA in some surrounding stromal cells, perpendicular or oblique processes originating from the stromal cells were observed penetrating and separating each neoplastic cell. Most of these penetrating processes expressed αSMA (Figure 9).

## 3. Discussion

In invasive lobular carcinoma of the breast, in which stromal cells express CD34 or αSMA around neoplastic cells, we demonstrate that (a) stromal cells adopt a similar arrangement and location to resident CD34+SCs/TCs in the normal breast, (b) CD34+ and αSMA+ stromal cells can coincide around the same nest of neoplastic cells and even around a single neoplastic cell and, that c) the stromal cells in the tumour coexpress CD34 and αSMA. The findings reveal that resident CD34+SCs/TCs participate as an important source of stromal cells (CAFs) in invasive lobular carcinoma. These issues are taken into consideration below.

Our observation of stromal cells expressing CD34 or αSMA around the strands and nests of neoplastic cells in invasive lobular carcinoma of the breast coincides with previous studies by other authors [34,42]. These studies included semiquantitative analysis of the differences in immunoreactivity for CD34 and αSMA in stromal cells and were mainly oriented to evaluating the prognostic relevance of the presence of one stromal cell type or another in the tumour [42]. For this purpose, the authors carried out a broad classification of the cases. With our objective in mind, we modified and simplified the classification and, depending on the predominant type/s of stromal cell/s in the tumour, grouped the cases as follows: (a) a predominance of CD34+SCs (26.01% of cases), (b) a predominance of αSMA+ stromal cells (30.9% of cases) and (c) a balanced presence of both markers (42.85% of cases).

To avoid confusing CD34+ ECs in cross sections of vessels with CD34+SCs and subsequently overestimating the latter, we considered the following: that ECs express CD34 marker more intensely than stromal cells; that αSMA+ pericytes are frequently observed even in smaller vessels when double staining with anti-CD34 and anti-αSMA is used; and above all that CD34 and CD31 are coexpressed in ECs but not in stromal cells [43,44]. When only stromal cells were immunostained, the nuclei of the neoplastic cells alone were revealed by the counterstain. The expression of cytokeratin AE1/AE3 was also verified in these neoplastic cells.

The flat (lamellar or velamentous) morphology of stromal cell processes in the invasive lobular carcinoma coincides with that of CD34+SCs/TCs in normal conditions. Indeed, other authors have used 3D imaging by FIB-SEM tomography to show telocyte telopodes as long, flattened, irregular veils [6,7,8]. We observed this fact in the stromal cells of tumours when the flat surface of the cell coincided with the section plane and in several individual visions at different heights using confocal microscopy. The coincidental presence of veils in telocytes and in stromal cells that surround neoplastic cells is a further unsought argument supporting the hypothesis of the participation of CD34+SCs/TCs in the origin of CAFs.

The correspondence in location and arrangement of the stromal cells surrounding neoplastic cells in invasive lobular carcinoma and the CD34+SCs/TCs observed by us and previously described by other authors in the normal breast [3,35,36,37,38,39,40,41] suggests that CD34+SCs/TCs act as a guide for neoplastic cells, condition specific morphological patterns and participate in the origin of stromal cells in the tumour. This corresponding circular arrangement of stromal cells around ducts supports the hypothesis of CD34+SCs as a source of CAFs and explains the presence of concentric rings of neoplastic cells around normal ducts. This finding is important because it is essentially unique for invasive lobular carcinoma in breast tumours and is therefore of diagnostic interest. Similar arrangements of CD34+SCs in the normal breast and of CAFs and neoplastic cells in other locations of the breast also explain other morphological patterns, such as around vessels and in adipose tissue and in smooth muscle of the nipple. Extensive tumoral regions in interlobular breast connective tissue, in which the neoplastic cell pathway of growth (generally in strands) is adapted to the longitudinal axis of the long, thin, veiled processes of stromal cells that partially preserve their normal arrangement, also strengthens the hypothesis that these cells originate from resident CD34+SCs/TCs and act as a guide for neoplastic cell growth.

The demonstration of CD34 and αSMA coexpression in stromal cells around neoplastic cells is the principal basis supporting the transit from CD34+SCs to αSMA+ stromal cells in invasive lobular carcinoma of the breast. A fact that supports the hypothesis on the participation of CD34+SCs/TCs in regenerative (local stem cell niche maintenance) and reparative (as progenitors) mechanisms [1,2,31,44,45,46,47,48,49]. Indeed, the presence of resident CD34+SFCs/TCs (there are no αSMA+ stromal cells) in the normal breast and of stromal cells that only express CD34, coexpress CD34 and αSMA, and only αSMA in the tumours strongly supports the origin in CD34+SCs/TCs resident in the normal breast of αSMA+ stromal cells in invasive lobular carcinoma. This origin is reinforced mainly by the same location and arrangement of stromal cells in the tumour as CD34+SCs/TCs in the normal breast. Stromal cells that delimit tumour nests and strands can form new processes that separate the neoplastic cells from each other. The fact that the newly formed process in stromal cells expresses αSMA is an indicator that changes in marker expression occur more quickly in cellular regions with a new adaptation.

Our study confirms that invasive lobular carcinoma is highly appropriate for studying CAFs. This suitability is mainly due to its growth in strands and small nests of neoplastic cells surrounded by stromal cells with a similar arrangement as in the normal breast and because marker expression is easy to follow in these stromal cells. The observation of an important location around vessels of CD34+ and αSMA+ stromal cells in the stroma of invasive lobular carcinoma is of interest, because the reactive microvasculature, which is conserved among tissues, participates in the evolution of reactive stroma (the reactive microvasculature hypothesis) [50]. This concept is mainly based on the recruitment of CD34+ fibroblasts in tumour associated reactive stroma and on the possibility that perivascular CD34+ stromal cells are progenitors of myofibroblasts [50].

We have focused our observations on investigating the origin of CAFs in invasive lobular carcinoma of the breast. Future studies in this type of tumour could confirm or contribute to several aspects of CAFs, such as CAF heterogeneity and plasticity, factors that participate in the conversion of CD34+SCs/TCs into CAFs, and CAF functions, including extracellular matrix deposition and remodelling, CAF-related factors that participate in tumour growth, invasion, metastasis and angiogenesis, as well as those that modulate tumour immunity (for review, see [29,51]). Future studies should also cover the co-localization of CD34 with other fibroblastic markers due to fibroblastic heterogeneity [52], including fibronectin, which promotes the eventual acquisition of a fibrotic response, and TGF β1, which activates the fibroinflammatory genomic program [53,54,55,56]. Likewise, the investigation of the origin of CAFs from CD34 + SCs/TCs should extend to tumors from other tissue locations (e.g., prostate, lung or skin), for which data that indicate this possibility have been provided [57,58,59].

In conclusion, our findings indicate that mammary resident CD34+SCs/TCs participate as a source of CAFs in invasive lobular carcinoma of the breast. The main findings are (a) the coincidental location and arrangement of normal breast CD34+SCs/TCs and tumour stromal cells, which may act as a guide for neoplastic cells, giving rise to specific morphological growth patterns, such as the typical concentric rings of neoplastic cells around normal ducts a fact of diagnostic interest, and (b) the presence of tumoral areas with stromal cells that coexpress CD34 and αSMA. Further studies on this issue are required in other types of epithelial tumours.

## 4. Material and Methods

### 4.1. Tissue Samples

The archives of Histology and Anatomical Pathology of the Department of Basic Medical Sciences of La Laguna University, University Hospital and Eurofins Megalab–Hospiten Hospitals of the Canary Islands were searched for cases with a diagnosis of invasive lobular carcinoma (classical morphological subtype) for the period January 2015 to July 2020, obtaining 42 cases. The patients were Caucasian, aged 35–70 years. Paraffin blocks were processed for histologic studies, including immunochemistry and immunofluorescence procedures. Specimens of six cases of normal breasts surgically resected for cosmetic reasons were used for the control. Ethical approval for this study was obtained from the Ethics Committee of La Laguna University CEIBA2021-0446, including the dissociation of the samples from any information that might identify the patients. The authors therefore had no access to identifiable patient information.

### 4.2. Light Microscopy

Specimens for conventional light microscopy were fixed in a buffered neutral 4% formaldehyde solution, embedded in paraffin and cut into 3 µm-thick sections. Sections were stained with Haematoxylin and Eosin (H&E).

### 4.3. Immunohistochemistry

Two types of procedures were performed. First procedure (automated): histologic sections, 3μm-thick, were attached to silanized slides. The sections were placed in a 60 °C oven overnight and in a BOND-MAX Automated Immunohistochemistry Vision Biosystem (Leica Microsystems GmbH, Wetzlar, Germany). Deparaffinized sections were pre-treated with the Epitope Retrieval Solution 2 (EDTA-buffer pH 8.8) at 98 °C for 20 min. Using the Bond Polymer Refine Detection Kit DC9800 (Leica Microsystems GmbH), peroxidase blocking was carried out for 10 min after washing steps. Newly washed, the tissues were incubated with anti-CD34 (Bond™ PA0212), anti-αSMA (Bond™ PA0943) or cytokeratin AE1/AE3 (Bond™ PA0094) primary antibody for 30 min, incubated with polymer for 10 min and developed with DAB Chromogen for 10 min. For double immunostaining (CD34/αSMA and CD34/cytokeratin), a similar automated procedure was performed with the following modification: (a) Bond™ Polymer Refine Detection Kit: chromogen DAB with the HRP enzyme which is visualised via a brown precipitate and (b) Bond™ Polymer Refine Red Detection Kit: chromogen Fast Red with the alkaline phosphatase (AP) enzyme which is visualized via a red precipitate. Second procedure (non-automated): histologic sections, 3 μm-thick, were attached to silanized slides. After rehydration, sections were boiled in 10 mM citrate buffer (pH 6) at 100 °C for 20 min for antigen retrieval, rinsed in Tris-buffered saline (TBS; pH 7.6, 0.05 M), blocked with 3% hydrogen peroxide and then incubated with the following primary antibodies diluted in TBS overnight in a humid chamber at room temperature: rabbit polyclonal anti-CD34 (1/100 dilution, code n°. A13929, AB clonal), mouse monoclonal anti αSMA (1/100 dilution, code n°. ABK1-A8914, Abyntek Biopharma) and mouse monoclonal anti-CD31 (Ready to use, code n°. IR610, clone JC70A, DAKO). The day after, sections were washed thrice in TBS and incubated for 1 h with biotinylated secondary goat anti-rabbit antibody (1/200, code n°. OS03B, Calbiochem) or biotinylated secondary goat anti-mouse antibody (1/200, code n°. 401213, Calbiochem), depending on the primary antibody. After several washes in TBS, sections were incubated for 1 h with streptavidin-biotin peroxidase complex (1/200, code n°. 189730, 2031941, Calbiochem) and developed using a TBS solution containing 0.04% 3,3′-diaminobenzidine (DAB) and 0.01% hydrogen peroxide by 5-min immersion. Sections were then briefly counterstained with Haematoxylin, dehydrated in ethanol series, cleared in xylene and mounted in Eukitt^®^. Positive and negative controls were used. For the double immunostaining, we used anti-CD34 antibody (diaminobenzidine, DAB, as chromogen) to highlight CD34+ ECs and stromal cells and anti-αSMA (aminoethylcarbazole, AEC, substrate-chromogen) for stromal cells and pericytes/smooth muscle cells.

### 4.4. Immunofluorescence in Confocal Microscopy

For immunofluorescence, tissue sections of 3 µm and 10 µm thick were obtained as described above. For antigen retrieval, sections were deparaffinized and boiled for 20 min in sodium citrate buffer 10 mM (pH 6), rinsed in Tris-buffered saline (TBS, pH 7.6, 0.05 M), and incubated with the following primary antibodies diluted in TBS overnight in a humid chamber at room temperature: rabbit polyclonal anti-CD34 (1/100 dilution, code n°. A13929, AB clonal), mouse monoclonal anti-CD31 (Ready to use, code n°. IR610, clone JC70A, DAKO) and mouse monoclonal anti-αSMA (1/100 dilution, code no. ABK1-A8914, Abyntek Biopharma). For double immunofluorescence staining, sections were incubated with a mixture of monoclonal and polyclonal primary antibodies. The next day, the slides were rinsed in TBS and incubated for one hour at room temperature in the dark with the secondary biotinylated goat anti-mouse IgG, H&L Chain specific Biotin conjugate (1:300, Calbiochem, cat. No. 401213, Calbiochem) and Alexa Fluor 488 goat anti-rabbit IgG (H+L) antibodies (1:300, cat. No. A11001, Invitrogen), followed by incubation with Streptavidin Cy3 conjugate (1:500, Code: SA1010, Invitrogen) for one hour at room temperature in the dark. Nuclei were detected by DAPI staining (Chemicon International, Temecula, CA, USA). After being washed in TBS, sections were exposed to a saturated solution of Sudan black B (Merck, Barcelona, Spain) for 20 min to block autofluorescence. They were rinsed in TBS and cover-slipped with DABCO (1%) and glycerol-PBS (1:1). Negative controls were performed in the absence of primary antibodies. Fluorescence immuno-signals were obtained using a Fluoview 1000 laser scanning confocal imaging system (Olympus Optical).

### 4.5. Electron Microscopy

Specimens of normal breast for electron microscopy were initially fixed in glutaraldehyde solution, diluted to 2% with sodium cacodylate buffer, pH 7.4, for six hours at 4 °C. They were then washed in the same buffer, post-fixed for two hours in 1% osmium tetroxide, dehydrated in a graded ethanol series, and embedded in epoxy resin. Ultrathin sections were double-stained with uranyl acetate and lead citrate. The grids were examined at 60 kV with a JEOL^®^ 100B Akishima (Tokyo, Japan) electron microscope.

### 4.6. Semiquantitative Analysis

The cases of invasive lobular carcinoma were grouped according to immunoreactivity for CD34 and αSMA in stromal cells. A semiquantitative analysis was carried out by two independent observers, following the procedure used with prognostic purposes by other authors [42] and modified and simplified by us. Thus, depending on the immunoreactivity of the stromal cells, the groups were (a) predominantly positive for CD34 (>70% in the tumour area), (b) balanced for CD34 and αSMA (30–70%) and (c) predominantly positive for αSMA (>70% in the tumour area). The percentages of the CD34 and αSMA stromal cells coinciding in single nest/strands and co-expressing CD34 and αSMA in the tumour areas with a higher incidence were considered in the aforementioned groups. Pearson’s chi-squared test was used for statistical analysis.

## Figures and Tables

**Figure 1 ijms-22-03686-f001:**
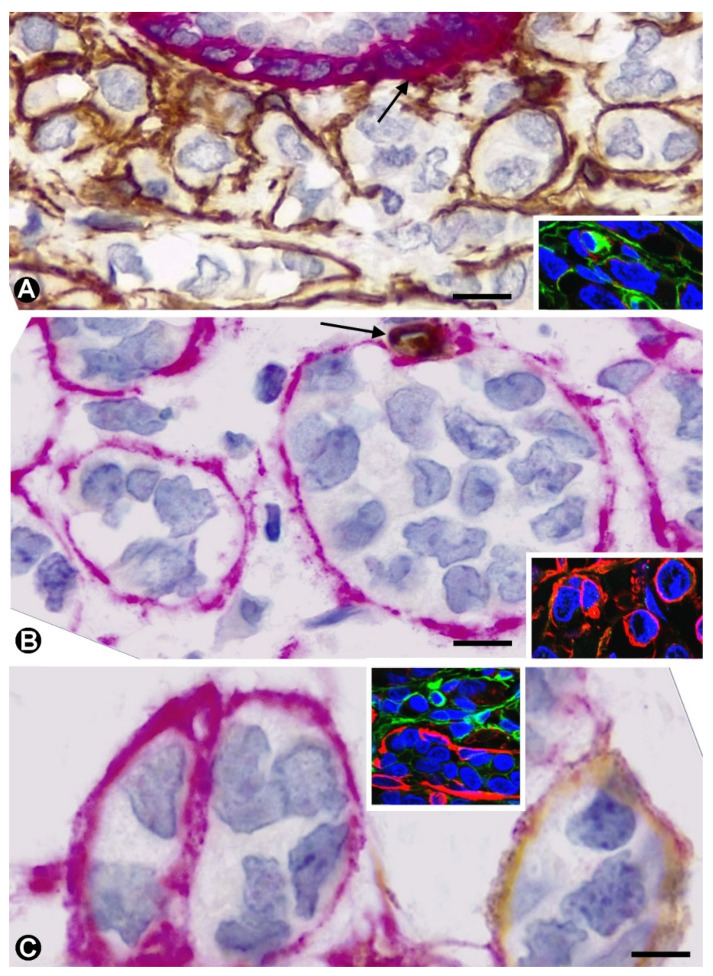
Stromal cells around nests of neoplastic cells in invasive lobular carcinoma of the breast. Double immunochemistry for CD34 (brown) and αSMA (red). Haematoxylin counterstain. Inserts: Double immunofluorescence labelling for CD34 (green) and αSMA (red) with DAPI (blue) counterstain. The stromal cells surrounding the neoplastic nests express CD34 (brown) in (**A**) and αSMA (red) in (**B**). In (**C**), the stromal cells express CD34 or αSMA, depending on the nest of neoplastic cells around which they are located. Note part of a duct in A, in which myoepithelial cells express αSMA (arrow). In B, a small vessel in which ECs express CD34 (arrow). The inserts show similar results by double immunofluorescence. Bar: (**A**): 15 µm; (**B**): 10 µm; (**C**): 8 µm.

**Figure 2 ijms-22-03686-f002:**
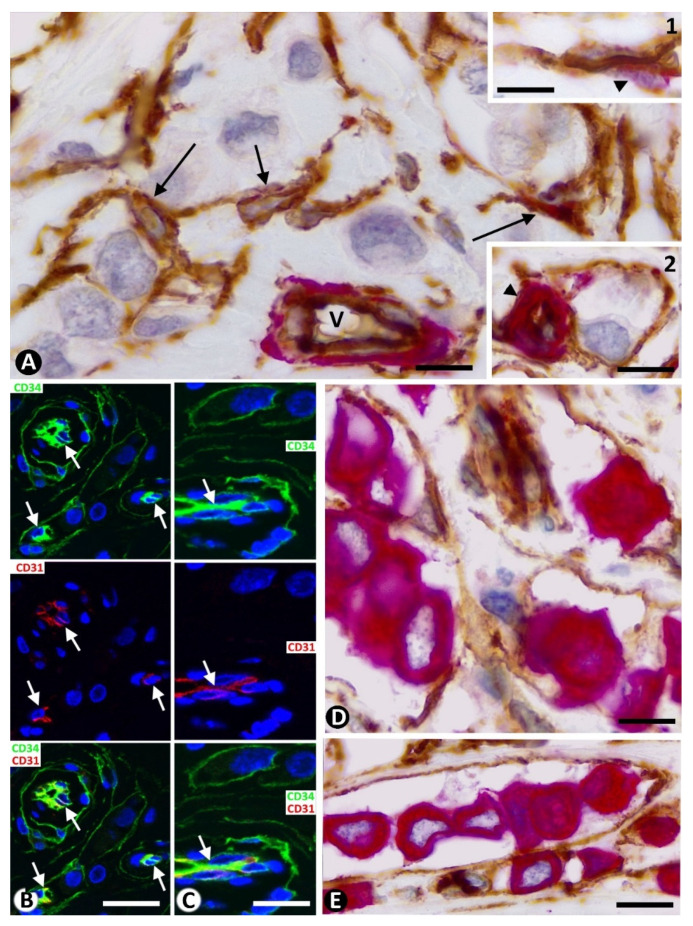
(**A**) and inserts: Double immunochemistry staining for CD34 (brown) and αSMA (red). Haematoxylin counterstain. Vessels (V) and stromal cells (arrows) are observed around neoplastic cells (Haematoxylin-stained nuclei). The expression of CD34 is more intense in ECs than in stromal cells. Pericytes (arrowhead) are observed around ECs (insert **1** and **2**). (**B**,**C**): Double immunofluorescence labelling for CD34 (green) and CD31 (red) with DAPI counterstain. Vessels (arrows), stromal cells (green) and nuclei of neoplastic cells are observed. Note CD34/CD31 coexpression in vessel ECs, while the stromal cells only express CD34. (**D**,**E**): Double immunochemistry staining for CD34 (brown) and Cytokeratin AE1/AE3 (red). Haematoxylin counterstain. Presence of neoplastic cells immunostained with anti-cytokeratin AE1/AE3 (red), surrounded by CD34+ stromal cells (brown). Bar: (**A**,**D**,**E**): 10 µm; (**B**,**C**): 30 µm; Insert **1** and **2** of (**A**): 20 µm.

**Figure 3 ijms-22-03686-f003:**
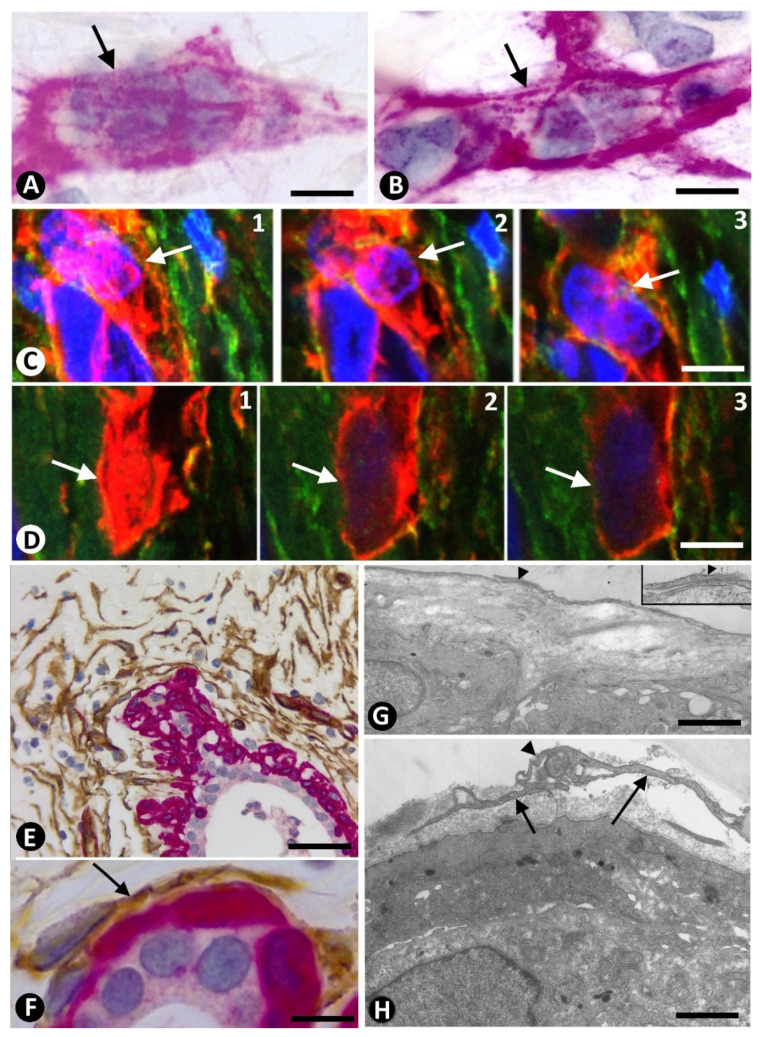
(**A**,**B**,**E**,**F**): Double immunochemistry for CD34 (brown) and αSMA (red). Haematoxylin counterstain. (**C**,**D**): Double immunofluorescence labelling for CD34 (green) and αSMA (red). DAPI counterstain. (**G**,**H**): Ultrathin sections. Uranyl acetate and Lead citrate. (**A**–**D**): Processes of αSMA+ stromal cells (red) forming veils (arrows) that cover neoplastic cells (blue stained nuclei) (in (**A**,**B**), sections parallel to the surface of the processes; in (**C**,**D**), individual visions at different heights in confocal microscopy, using tissue sections of 10 µm). (**E**): Several layers of CD34+SCs/TCs (brown) around a normal breast duct in which the external layer of myoepithelial cells expresses αSMA (red). (**F**): One CD34+SC (brown, arrow) is observed around myoepithelial cells (red) of a component of the terminal ductal-lobular unit. (**G**,**H**): Ultrastructural images of long, thin telocyte extensions (telopodes) with podomeres (H arrows) and podoms (arrowhead) around ductal myoepithelial cells. Insert of (**G**) shows a contact between telopodes (arrowhead). Bar: (**A**–**D**): 8 µm; (**E**): 30 µm; (**F**): 20 µm; (**G**,**H**): 4 µm.

**Figure 4 ijms-22-03686-f004:**
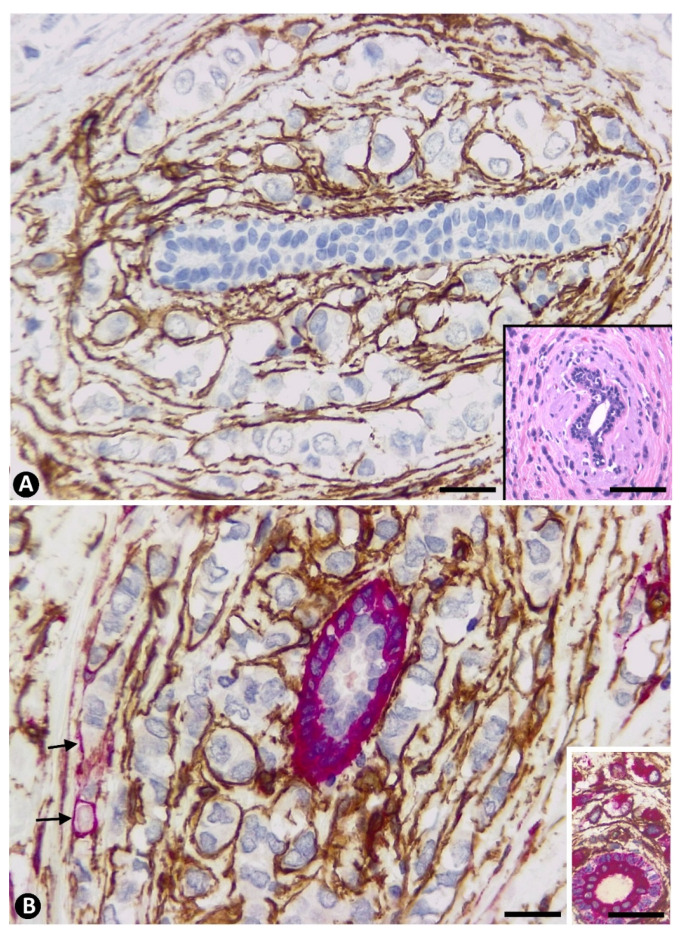
(**A**): Immunochemistry staining for CD34. Haematoxylin counterstain. Rings of CD34+SCs (brown) surrounding neoplastic cells around a glandular duct. Insert: a duct with the peripheral rings of neoplastic cells, which is virtually diagnostic of invasive lobular carcinoma in Haematoxylin-eosin staining. (**B**): Double immunochemistry staining for CD34 (brown) and αSMA (red). Haematoxylin counterstain. Abundant CD34+ (brown) and scarce SMA+ (red, arrows) stromal cells around a duct with αSMA+ myoepithelial cells. Insert: Double immunochemistry staining for CD34 (brown) and cytokeratin AE1/AE3 (red). Epithelial cells of the duct and neoplastic cells express cytokeratin AE1/AE3 (red) and stromal cells express CD34 (brown). Bar: (**A**,**B**): 30 µm; Insert of (**A**): 120 µm and Insert of (**B**): 80 µm.

**Figure 5 ijms-22-03686-f005:**
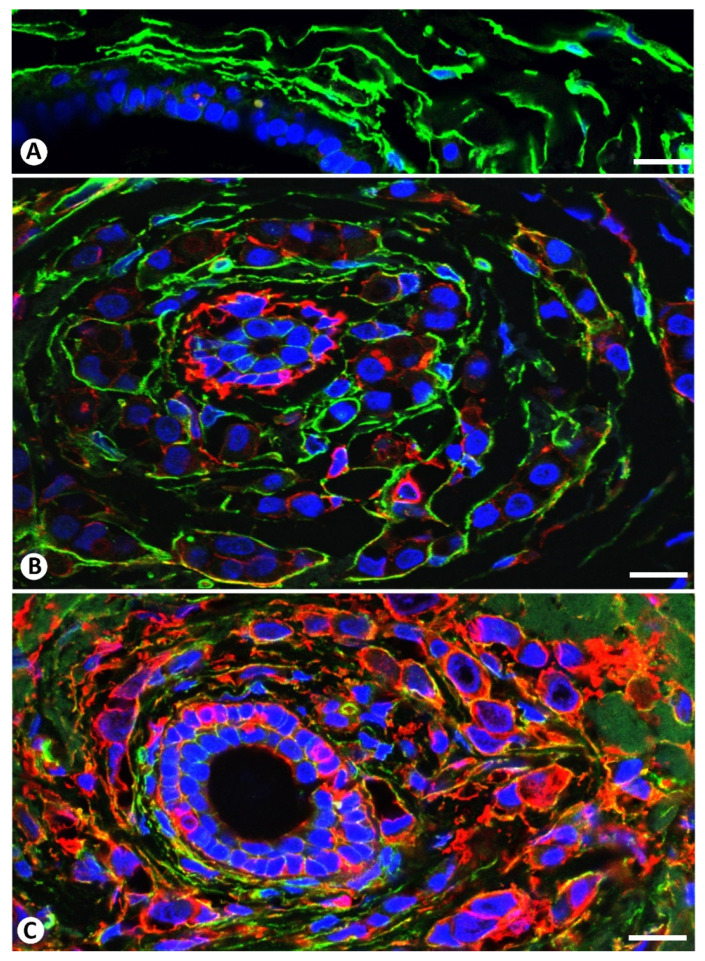
Double immunofluorescence labelling for CD34 (green) and αSMA (red) with DAPI (blue) counterstain. (**A**): CD34+SCs/TCs (green) around a duct. (**B**,**C**): Rings of CD34+ (green) and αSMA+ (red) stromal cells surrounding neoplastic cells (blue stained nuclei) around ducts. Note the predominance of CD34+ stromal cells in (**B**) and of αSMA+ stromal cells in (**C**). Some stromal cells coexpress both markers. Bar: (**A**): 40 µm; (**B**,**C**): 30 µm.

**Figure 6 ijms-22-03686-f006:**
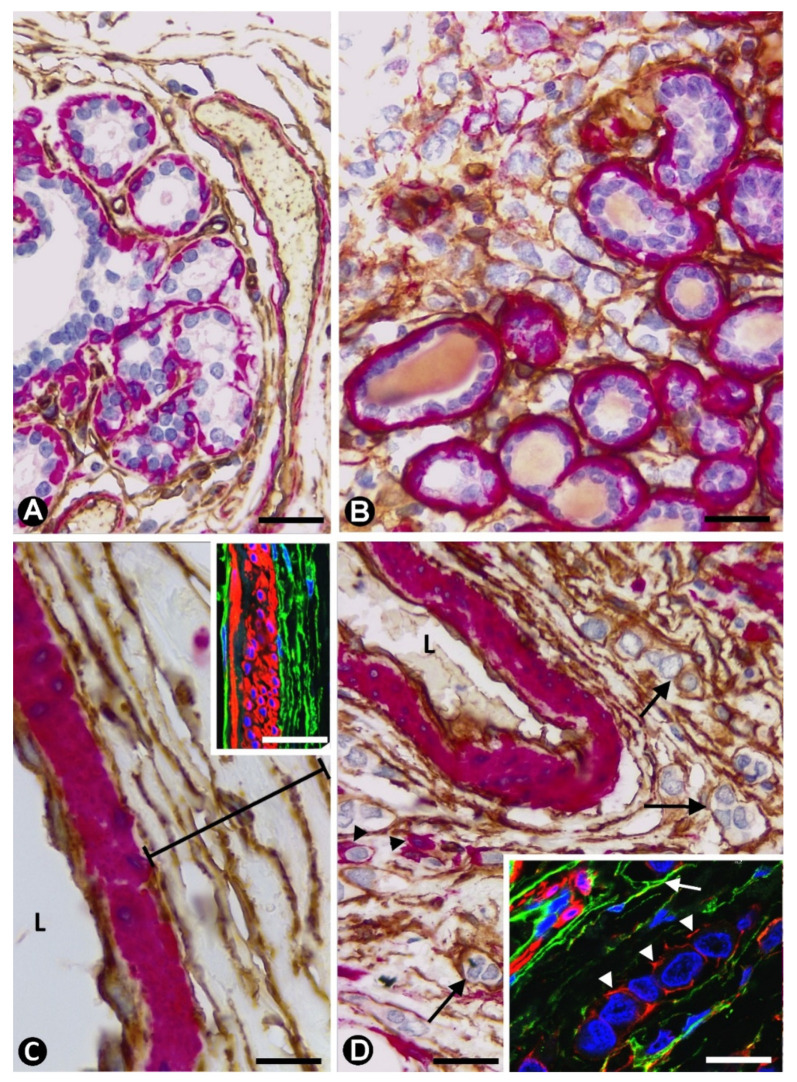
Double immunochemistry for CD34 (brown) and αSMA (red). Haematoxylin counterstain. (**A**): Part of a lobule in which CD34+SCs/TCs are observed in the interstitium. (**B**): CD34+ stromal cells surrounding neoplastic cells in the interstitium of a lobule. (**C**): Several layers of CD34+SCs/TCs are seen in the adventitia of an artery. The insert shows a similar image in double immunofluorescence labelling. (**D**): CD34+ stromal cells (arrows) and occasional αSMA+ cells (arrowhead) around neoplastic cells in the adventitia of an artery. The insert shows positivity for CD34 (arrow) and αSMA (arrowheads) in stromal cells. Bar: (**A**,**B**,**D**): 30 µm; C: 15 µm; Insert of (**C**,**D**): 60 µm.

**Figure 7 ijms-22-03686-f007:**
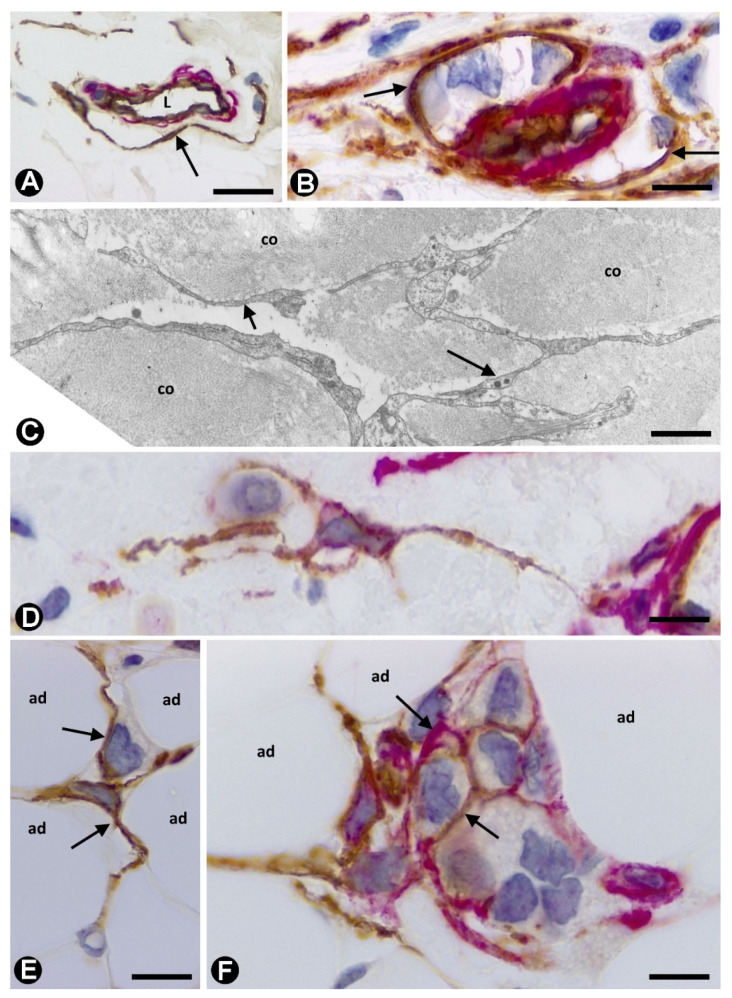
(**A**,**B**,**D**–**F**): Double immunochemistry staining for CD34 (brown) and αSMA (red). Haematoxylin counterstain. (**C**): Ultrathin section. Uranyl acetate and Lead citrate. (**A**): One CD34+SC/TC (arrow) around a small vessel. (**B**): Stromal cell processes (arrows) are observed surrounding neoplastic cells around a small vessel. (**C**): Ultrastructural image of telocyte telopodes (arrows) between collagen (CO) components in the interlobular connective tissue. (**D**): One CD34+ stromal cell with long processes, one of which surrounds a neoplastic cell. (**E**): CD34+SCs/TCs (arrows) between adipocytes (ad). (**F**): CD34+ (brown) and αSMA+ (red) stromal cells (arrows) are seen around a nest of neoplastic cells between adipocytes (ad). Bar: (**A**): 30 µm; (**B**,**F**): 20 µm; (**C**): 4 µm; (**D**,**E**): 15 µm.

**Figure 8 ijms-22-03686-f008:**
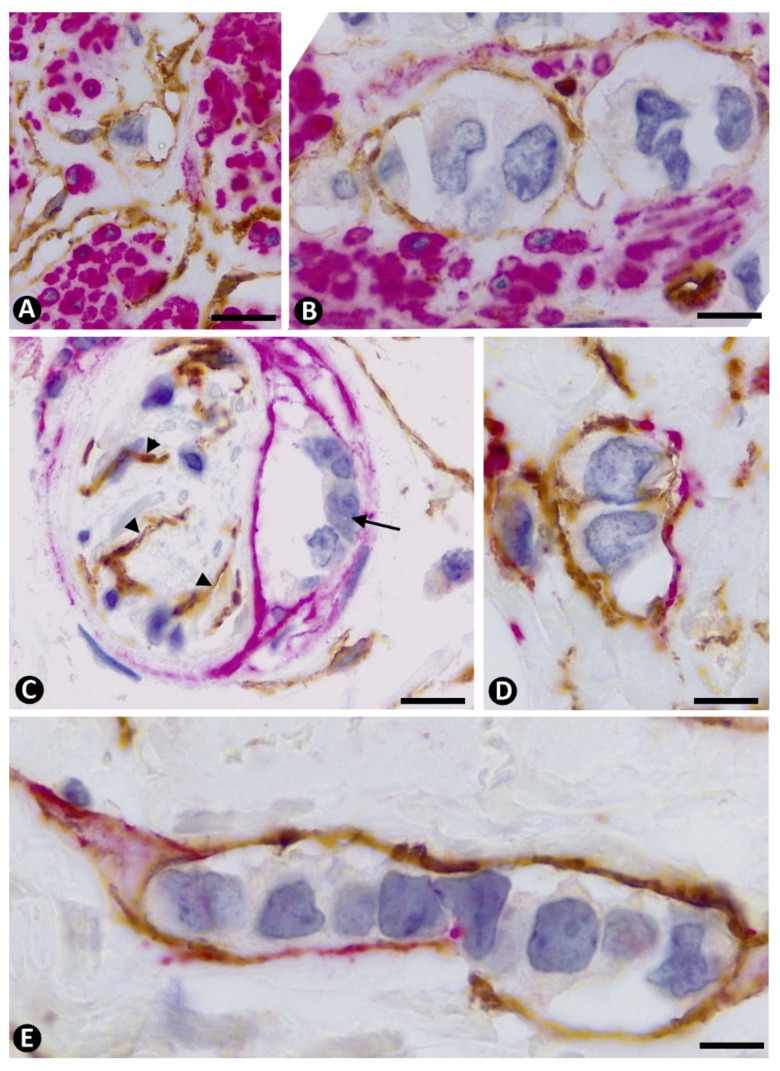
Double immunochemistry labelling for CD34 (brown) and αSMA (red). Haematoxylin counterstain. (**A**,**B**): CD34+SCs (brown) around neoplastic cells in the smooth muscle fascicles (red) of the breast nipple. (**C**): αSMA+ stromal cells (red) in the epi-perineurium of a small nerve surrounding neoplastic cells (arrow). Note CD34+SCs/TCs (brown, arrowheads) in the endoneurium. (**D**,**E**): CD34+ (brown) and αSMA+ (red) stromal cells around the same nest (**D**) and strand (**E**) of neoplastic cells. Bar: (**A**): 30 µm; (**B**,**D**): 10 µm; (**C**) 25 µm; (**E**): 10 µm.

**Figure 9 ijms-22-03686-f009:**
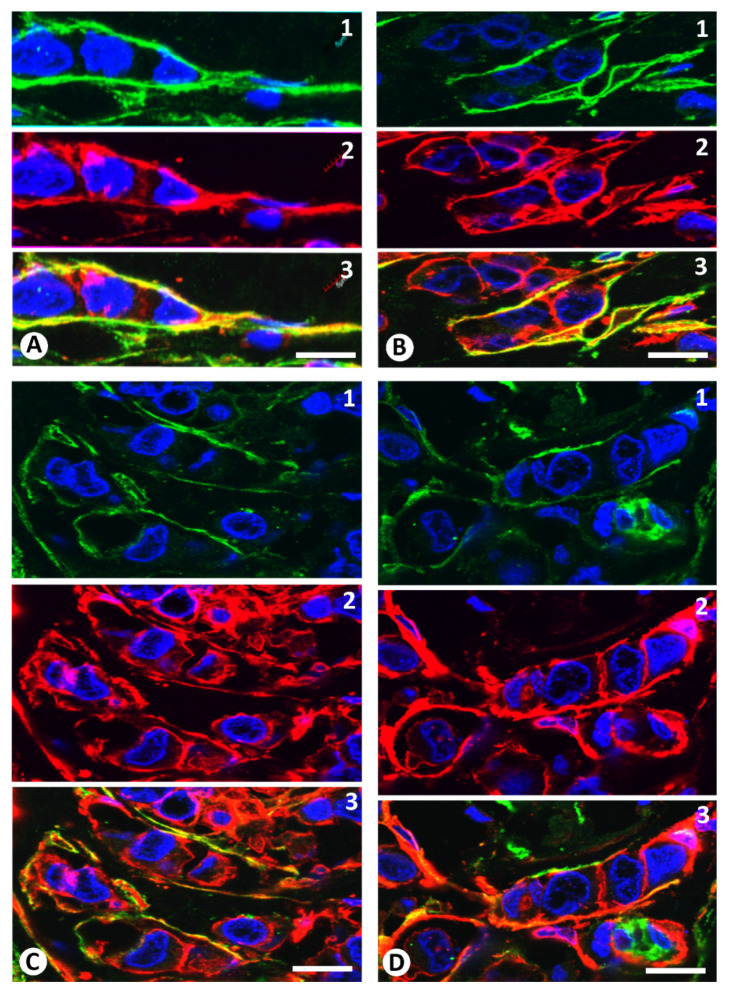
Double immunofluorescence labelling for CD34 (green) and αSMA (red) with DAPI (blue) counterstain. Coexpression of CD34 and αSMA is observed in stromal cells around neoplastic cells. Note cells expressing CD34 (green, (**A**)**1**, (**B**)**1**, (**C**)**1** and (**D**)**1**), αSMA (red, (**A**)**2**, (**B**)**2**, (**C**)**2** and (**D**)**2**) and both markers merged (yellow, (**A**)**3**, (**B**)**3**, (**C**)**3** and (**D**)**3**). Bar: (**A**,**B**,**D**): 10 µm; (**C**): 20 µm.

**Figure 10 ijms-22-03686-f010:**
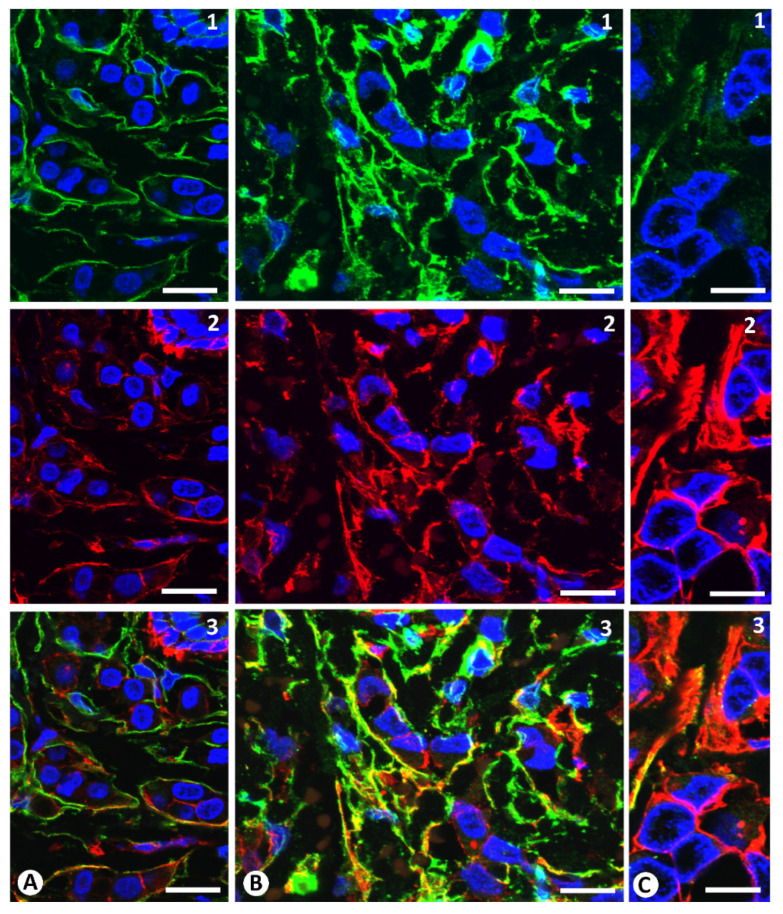
Double immunofluorescence labelling for CD34 (green) and αSMA (red) with DAPI (blue) counterstain. Numerous cells expressing both markers are observed where CD34+SCs predominate (**A**) or where there is a balanced proportion of CD34+ and αSMA+ stromal cells (**B**) rather than where αSMA+ stromal cells predominate (**C**). Numbers indicate the expression of CD34 (**1**), αSMA (**2**) and merged expression of CD34 and αSMA (**3**). Bar: (**A**–**C**): 20 µm.

**Table 1 ijms-22-03686-t001:** Numbers and percentage of cases, and percentage of stromal cells expressing CD34 and αSMA in normal breast (control cases: 6) and in tumor stroma of invasive lobular carcinoma of the breast (total cases: 42) grouped depending on the predominance or balanced proportion of each stromal cell type.

	Normal Breast			Invasive Lobular Carcinoma of the Breast		
	N° and Percentage of Cases	Percentage of CD34+SCs/TCs	Percentage of αSMA+ Stromal Cells	N° and Percentage of Cases	Percentage of CD34+SCs/TCs	Percentage of αSMA+ Stromal Cells
Predominance of CD34+SCs/TCs	6 (100%)	100%	0%	11 (26.19%)	>70%	<30%
Balanced Proportion of CD34+ and αSMA Stromal Cells	0%	0%	0%	18 (42.85%)	30–70%	30–70%
Predominance of αSMA+ Stromal Cells	0%	0%	0%	13 (30.95%)	<30%	>70%

**Table 2 ijms-22-03686-t002:** Correspondence in location and arrangement of CD34+ and SMA+ stromal cells in the normal breast and in the stroma of invasive lobular carcinoma, the percentages of both type of cells coinciding around a same nest/strand of neoplastic cells, and the percentages of cells co-expressing both markers (*p* < 0.002).

	Normal Breast		Invasive Lobular Carcinoma		
	CD34+ Stromal Cells	αSMA+ Stromal Cells	Predominance of CD34+ Stromal Cells	Balanced Proportion of CD34+ and αSMA+ Stromal Cells	Predominance of αSMA+ Stromal Cells
N° and %	6 (100%)	0 (0%)	11 (26.19%)	18 (42.85%)	13 (30.95%)
Correspondence in Location and Arrangement of CD34+ and αSMA+ Stromal Cells	Periductal, Perivascular, Interlobular interstitium, septal and intralobular adipose tissue smooth muscle of the nipples		Coincidence with normal breast	Coincidence with normal breast	Coincidence with normal breast
Coincidence of Both Types of Stromal Cells in Neoplastic Nests/Strands	0	0	26.09 ± 4.39	32,27 ± 8.44	22.07 ± 5.23
Coexpression of CD34+ and αSMA+ in Stromal Cells (mean ± s.d.)	0	0	27.63 ± 3.11	29.77 ± 6.50	25.25 ± 4.28

## Data Availability

All the data are reported in the present paper.

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
