# Peer review of "CD34+ Stromal Cells/Telocytes as a Source of Cancer-Associated Fibroblasts (CAFs) in Invasive Lobular Carcinoma of the Breast"

_ijms, 2021, doi:10.3390/ijms22073686_

Round 1

Reviewer 1 Report

In my opinion paper have good quality, but there should be presented some tables including percentage of positive stained cases of each marker. The statistical analysis of presented results also should be provided. Authors should also consider to make some comparing with control tissue. 

Author Response

Tables showing the percentage of positive stained cases of each marker, the coincidence of CD34+ and SMA+ stromal cells around single nests/strands of neoplastic cells, and the percentage of cells expressing both markers have been provided. Comparison with control tissue has also been undertaken. Thus, Table 1 includes the numbers and percentage of cases and the percentages of CD34+ and SMA+ stromal cells in the normal breast and in invasive lobular carcinoma grouped depending on the predominance or balanced proportion of the cells expressing the markers. In Table 2, we present the coincidence in location and arrangement of CD34+SCs/TCs in the normal breast and of CD34 and SMA+ stromal cells in the stroma of invasive lobular carcinoma, the percentages of both types of cells coinciding around single nests/strands of neoplastic cells, and the percentages of cells co-expressing both markers. Statistical analysis has also been carried out.  

Thank you for improving this manuscript.

Reviewer 2 Report

Diaz-Flores et al presented studies on the role of cancer-associated fibroblasts (CAFs) in breast cancer. It is an emerging topic in cancer field to target CAFs for better outcome in cancer prevention and treatment. Here, the authors focus on the expression of CD34 in CAFs which is not entirely novel in the field. It has been previously published from another group and yet not being cited in this paper (DOI: 10.1016/j.ajpath.2014.02.021  or https://pubmed.ncbi.nlm.nih.gov/24713391/). Additionally, it is difficult to make any conclusions based on the results, especially there is no single quantification has been done in the paper only representative images.    Additionally, the authors have not address the functional relevance of CD34 or αSMA around neoplastic cells. e.g., by depleting the expression of these markers in vitro or in vivo. The authors should also present the correlation of high or low CD34 & αSMA expression to overall patient survival. Finally, the authors should also verify co-localisation of CD34 with other fibroblast markers such as fibronectin or TGF-beta1 since not all CAFs express aSMA ( DOI: 10.1016/j.ccell.2018.01.011). Therefore, I would not recommend publication of this research article in IJMS due to the lack of novelty and poor presentation of results.

Author Response

We now refer to the concepts and authors mentioned and have expanded the information on the semi-quantitative studies.

In the discussion, we include the following paragraph: “The observation of an important location around vessels of CD34+ and αSMA+ stromal cells in the stroma of invasive lobular carcinoma is interesting because the reactive microvasculature, which is conserved among tissues, participates in the evolution of reactive stroma (the reactive microvasculature hypothesis) [50]. This concept is mainly based on the recruitment of CD34+ fibroblasts in tumour associated reactive stroma and on the possibility that perivascular CD34+ stromal cells are progenitors of myofibroblasts [50].”

The expanded semi-quantitative studies are presented in tables showing the percentage of positive stained cases of each marker, the coincidence of CD34+ and αSMA+ stromal cells around single nests/strands of neoplastic cells, and the percentage of cells expressing both markers. Comparison with control tissue has also been undertaken. Thus, Table 1 includes the numbers and percentage of cases and the percentages of CD34+ and αSMA+ stromal cells in the normal breast and in invasive lobular carcinoma grouped depending on the predominance or balanced proportion of the cells expressing the markers. In Table 2, we present the coincidence in location and arrangement of CD34+SCs/TCs in the normal breast and of CD34 and αSMA+ stromal cells in the stroma of invasive lobular carcinoma, the percentages of both types of cells coinciding around single nests/strands of neoplastic cells, and the percentages of cells co-expressing both markers. Statistical analysis has also been carried out.  

The new patient information now includes the data of when the cases were obtained (January 2015–July 2020). Unfortunately, we cannot establish overall patient survival.

In Discussion we included a new paragraph: “Future studies should also include the co-localization of CD34 with other fibroblastic markers due to fibroblastic heterogeneity [52], including fibronectin, which promotes the eventual acquisition of a fibrotic response, and TGF β1, which activates fibroinflammatory genomic program [53-56].”

We have focused our observations on investigating the origin of CAFs from CD34+SCs/TCs in invasive lobular carcinoma considering that the arrangement and characteristics of CD34+SCs/TCs have been well typified in the normal breast and that both CD34+ and αSMA+ stromal cells are present in the tumour stroma, acquiring a specific distribution. These properties have allowed us to contribute fats in the tumour itself (including coexpression of markers), which supports the hypothesis on the participation of CD34+SCs/TCs in CAFs. We hope to have complied with your observations to the extent that we could.

Thank you for improving this manuscript. 

Round 2

Reviewer 1 Report

Paper could accepted for printing in this form.

Author Response

Thank you very much

Reviewer 2 Report

The authors have addressed my comments sufficiently with addition of quantification and further clarification on the result section. In addition, I would recommend to include some information that it will be interesting in the future to validate the findings in CAFs from other tissue origin, in particularly those that have enriched signatures mentioned in this study, such as in prostate (https://doi.org/10.1186/s13046-020-1542-z), in lung (https://doi.org/10.1111/1759-7714.13248), and in skin (https://doi.org/10.15252/emmm.201911466).

Author Response

Following your indications, we add in the Discussion: “Likewise, the investigation of the origin of CAFs from CD34 + SCs/TCs should extend to tumors from other tissue locations (e.g. prostate, lung or skin), for which data that indicate this possibility have been provided [57-59].”

Thank you for improving this manuscript.